# Mechanically Strong, Liquid-Resistant Photothermal Bioplastic Constructed from Cellulose and Metal-Organic Framework for Light-Driven Mechanical Motion

**DOI:** 10.3390/molecules26154449

**Published:** 2021-07-23

**Authors:** Lijian Sun, Limei Li, Xianhui An, Xueren Qian

**Affiliations:** Key Laboratory of Bio-Based Material Science & Technology, Northeast Forestry University, Ministry of Education, Harbin 150040, China; sunlijian0308@163.com (L.S.); limeili937@foxmail.com (L.L.); anxianh509@163.com (X.A.)

**Keywords:** bioplastic, Prussian blue, liquid-resistant, photothermal conversion, mechanical motion, mechanical properties

## Abstract

The development of photothermal materials with a high light-to-heat conversion capability is essential for the utilization of clean solar energy. In this work, we demonstrate the use of a novel and sustainable concept involving cellulose liquefaction, rapid gelation, in situ synthesis and hot-press drying to convert cellulose and metal–organic framework (Prussian blue) into a stable photothermal bioplastic that can harvest sunlight and convert it into mechanical motion. As expected, the obtained Prussian blue@cellulose bioplastic (PCBP) can effectively absorb sunlight and the surface can be heated up to 70.3 °C under one sun irradiation (100 mW cm^−2^). As a demonstration of the practicality of PCBP, it was successfully used to drive a Stirling engine motion. Meanwhile, hot-pressing promotes the densification of the structure of PCBP and, therefore, improves the resistance to the penetration of water/non-aqueous liquids. Moreover, PCBP shows good mechanical properties and thermal stability. Given the excellent photothermal performance and environmentally friendly features of photothermal conversion bioplastic, we envisage this sustainable plastic film could play important roles toward diversified applications: a photothermal layer for thermoelectric generator, agricultural films for soil mulching and photothermal antibacterial activity, among others.

## 1. Introduction

In academia and the industry, the conversion of electricity, chemical energy, fossil fuels, light energy and other forms of energy sources into mechanical motion has attracted widespread attention. As a sustainable, inexhaustible source of clean energy, solar energy offers an effective solution to alleviate the energy crisis [1,2]. Therefore, the conversion of solar energy to mechanical motion is particularly attractive. The effective way to realize the conversion of solar energy into mechanical motion is to convert solar energy into thermal energy, which in turn is converted into mechanical motion. Photothermal conversion materials could achieve sequential energy conversion by effectively harvesting solar energy. Some photothermal nanomaterials have been developed, including carbon-based materials [3,4,5], noble-metal nanoparticles [6,7,8,9], black inorganic semiconductors [10,11,12] and low-cost conductive polymers [13,14]. Meanwhile, many organic–inorganic composite materials, such as MoS_2_/SWNTs [15], Al-Ti-O/polyvinylidene fluoride [16], Ti_2_O_3_/cellulose [17], etc., have also been developed for use in photothermal materials. However, most of the previous research of photothermal materials have focused on tumor photothermal therapy [18], photothermal antibacterial activity [19], wastewater purification and desalination [20,21], but little research has been conducted on the conversion of light energy to mechanical motion [22,23].

Metal–organic frameworks (MOFs) are emerging porous crystalline materials that have attracted wide attention due to their wide range of applications [24,25,26]. To the best of our knowledge, the use of MOFs in solar thermal materials is rarely reported. Recently, Prussian blue (PB) particles have attracted much attention owing to their biosafety, biocompatibility, green and simple synthesis, low cost, high stability, adjustable morphology, and excellent photothermal performance [27,28,29,30]. PB belongs to a typical transition metal–organic framework (MOF), which can be fabricated by simple co-precipitation of [Fe(CN)_6_]^4−^ and Fe^3+^ [31]. PB has a remarkable photothermal effect due to the charge transfer between the metal from Fe^2+^ to Fe^3+^ [32,33,34], which has a light-to-heat conversion efficiency higher than or equal to that of carbon nanomaterials and noble Au [35,36], but it involves a far lower cost. Compared with other MOFs, PB has attractive water and organic solvent stability, especially its excellent resistance to organic solvents making it suitable for a wide range of solvent media [37]. Therefore, in this work, PB is reasonably selected as a photothermal material in consideration of its excellent photothermal effect, low cost and easy synthesis characteristics. However, the disadvantageous nature of PB, such as inherent fragility and unsatisfied processability, hinder some practical applications. Therefore, it is urgent to select a suitable and flexible substrate to solve these limitations.

Cellulose shows great promise for use as a supporting substrate or template material to PB due to renewability, degradability, abundance in storage and low cost [38]. In fact, numerous MOFs/cellulose composite materials have been successfully developed and applied in various fields such as antibacterial [39], UV shielding [40], catalysis [41], flame retardant [42], fluorescence [43], metal ion adsorption [44] and supercapacitor [45], etc. Inspired by these works, we attempt to combine photothermal PB with a natural polymer, i.e., cellulose, to prepare bioplastic with excellent photothermal effect, liquid resistance and high mechanical strength. Using this photothermal cellulose-based bioplastic, solar energy can be harvested and converted into mechanical motion. Due to the rapid growth in demand for environmentally friendly materials and green energy, it makes sense to manufacture a new type of green plastic with excellent photothermal performance.

In the present work, PB particles are in situ loaded into regenerated cellulose hydrogel and then hot pressed to form photothermal bioplastics. The as-prepared photothermal bioplastics have good mechanical properties, flexibility and efficient photothermal conversion performance. The obtained photothermal bioplastic can effectively absorb sunlight and convert it to heat, and the surface of bioplastic can be heated to 70.3 °C under one sunlight. Encouraged by this result, we used the as-fabricated photothermal bioplastic to drive the Stirling engine motion to realize the goal of “solar light to mechanical motion”. The strong barrier of bioplastic to penetration by aqueous/nonaqueous liquids was demonstrated. Our proposed strategy is simple, green and easy to operate both in the work-up and purification stage, in accordance with principles of green chemistry and sustainable development. The fabrication process was carried out in an aqueous medium without any other toxic solvents. This simple and sustainable strategy would direct green photothermal conversion bioplastic toward diversified applications: photothermal layer for thermoelectric generator, agricultural films for soil mulching and photothermal antibacterial action, among others.

## 2. Results

### 2.1. The Formation of Prussian Blue@Cellulose Bioplastic (PCBP)

Scheme 1 schematically illustrates the preparation of PCBP. Initially, the synthesis process involved the preparation of a homogeneous cellulose/LiOH/urea aqueous solution, followed by a physical gelation with anhydrous ethanol and washed with water to form a cellulose hydrogel. The cellulose hydrogel with a thickness of about 2.0 mm in Figure 1a is transparent. Subsequently, PB particles were in situ loaded into the regenerated cellulose hydrogel to obtain a PB@cellulose composite hydrogel. Finally, the PB@cellulose composite hydrogel was hot-pressed to obtain composite bioplastic. The hot-pressing not only rapidly evaporated and removed the water from the cellulose hydrogel, but also changed the orientation and crystalline structure of the cellulose, leading to the structural densification. In Figure 1b, the obtained pure cellulose bioplastic (CBP) with about 0.2 mm thickness is transparent and shows a relatively high transmittance (85.1% at 600 nm; Appendix A). As shown in Figure 1c,d, the PB @cellulose composite hydrogel and PCBP exhibit a dark blue color, indicating the accumulation of PB particles inside or on the surface of the PB@cellulose composite hydrogel.

In this study, PB with cubic crystals was synthesized from a single iron-source Na_4_Fe(CN)_6_. Appendix A show the SEM images and crystal structure of PB. To obtain the morphology of the functional bioplastic filled with PB particles, a representative cross-sectional SEM image of CBP and PCBP-1 was observed. It can clearly be seen in Appendix A and that cellulose microfibrils show a parallel arrangement, demonstrating that the cellulose hydrogel composed of cellulose molecular chains was transferred into transparent CBP after hot pressing. In Figure 2, we can see the existence of PB particles in the network structure of PCBP-1, the PB particles are tightly encapsulated in the PCBP-1 with the magnified SEM image, the PB particles (circled) are well-distributed in the PCBP-1 without congregation. PB particles are firmly embedded in the cellulose matrix, which indicates a strong interfacial adhesion between cellulose and PB particles. The uniform distribution and the alignment of PB particles, as well as a strong interface, are conducive to improving the light absorption and heat conduction. An EDS analysis also confirmed the formation of PB particles inside PCBP (Figure 2c). Signals of nitrogen, iron and sodium were recorded in EDS analysis patterns for cellulose decorated by PB. Elemental mapping images (Figure 2d) of PCBP-1 also revealed the uniform distributing of PB particles in PCBP-1, as demonstrated by the uniform detection of iron, sodium and nitrogen atoms in PCBP-1 apart from the carbon and oxygen atoms.

The surface SEM images of the CBP, PCBP-1 and PCBP-2 are displayed in Figure 3. For comparison purposes, the SEM images of the surface morphology of the CBP are shown in Figure 3a–c. The surface of CBP shows desirable flatness, homogeneity and high compactness, which is attributed to plastic deformation after hot pressing. The PB particles are uniformly dispersed on the surface of PCBP without visible aggregates shown in the SEM images of PCBP-1 and PCBP-2. It can be seen in Figure 3d–f that some PB particles on the surface of PCBP-1 were trapped in the cellulose matrix. This is because the PB particles are closely attached to the cellulose due to hot pressing. This result indicates that PB was not only formed inside the hydrogel, but also a large amount of PB was formed on the surface of the hydrogel. Compared with PCBP-1, PCBP-2 had more PB particles on its surface (Figure 3g–i). It can be seen that the PB particles are embedded in the cellulose matrix.

To confirm the existence of PB particles in the prepared photothermal bioplastic, the chemical structure of PB, pure CBP and PCBP-1 was studied by FTIR, as shown in Figure 4a. In the FTIR spectrum of the PB, the peaks at 2086 and 601 cm^−1^ were attributed to the C=N stretching and Fe–O formation [46], respectively. For neat CBP, the characteristic peaks at 3321 and 2888 cm^−1^ were ascribed to the O–H and C–H stretching vibrations of the sugar ring, respectively. The peaks at 1644 and 1021 cm^−1^ were ascribed to the stretching vibration of C=O and C–O. The same peaks were observed for the PCBP-1. After introduction of PB, the occurrence of a new peak at 2086 cm^−1^ attributed to the vibration of C=N in PCBP-1, demonstrating successful synthesis of PB in PCBP-1.

The formation of PB in the photothermal bioplastic was further confirmed by using an XRD analysis of CBP, PB and PCBP powders. Dark blue PB particles were fabricated from a single iron-source Na_4_Fe(CN)_6_. Figure 4b shows that the XRD diffraction spectrum of the CBP had three strong diffraction peaks at 12.4°, 20.2°, and 22.2°, corresponding to the diffraction of the (1–10), (110) and (200) planes of the cellulose II-type, demonstrating that the crystal structure of the sample did not change [47]. For pure PB powder, diffraction peaks at 2θ° = 57.1°, 54.0°, 50.7°, 43.5°, 39.5°, 35.5°, 24.9°, and 17.5° corresponded to the (620), (600), (440), (422), (420), (400), (220) and (200) planes, respectively [27]. The PCBP also showed diffraction peaks at the (620), (600), (440), (422), (420), (400), (220) and (200) planes, which are attributed to the successful synthesis of PB particles into the cellulose network.

As shown in Figure 4c, the CBP showed a three-step thermal degradation behavior: The first step corresponding to a small mass loss appeared at 75 °C. The subsequent mass loss in the range of 150–210 °C is attributed to the evaporation of residual moisture from the specimens. The next mass loss in the range of 270–350 °C, was the sign of carbohydrate polymer degradation. The final 10% mass loss at 370–800 °C could be ascribed to the oxidation of the residues, leaving a little amount of residue (13.7 wt%) at 800 °C. Similar to CBP, the decomposition of PCBP-1 and PCBP-2 also involved three similar steps. It should be noted that the thermal stability of PCBP-1 and PCBP-2 increased and they had higher amounts of the residual chars. The mentioned enhancement could be due to the existence of PB, as it efficiently prevents the volatilization of the decomposition products into the gas phase.

The chemical composition and electronic structure of the CBP and PCBP-1 were characterized by XPS measurements. As seen in the wide-scan XPS spectra of PB, iron (Fe 2p), carbon (C 1s), oxygen (O 1s) and nitrogen (N 1s), signal peaks were present at around 708.4, 284.6, 532.7 and 399.6 eV, respectively, suggesting Fe, C, O and N four elements existing on PB. The neat CBP consisted of (β1→4)-linked D-glucose units containing only C 1s and O 1s peaks at 284.8 and 533.1 eV, and no N 1s and Fe 2p peaks. (Figure 4d). The XPS survey scan of the PCBP-1 showed C 1s, N 1s, O 1s, and Fe 2p peaks with binding energies of 284.6, 399.6, 532.7 and 708.4 eV, respectively. The O 1s spectrum (Figure 4f) of the PCBP-1 showed two peaks at 531.21 and 286.56 eV, corresponding to Fe–O and C–O, respectively. The spectrum clearly showed a chemical bond between cellulose and PB based on the existence of the Fe–O peak [48,49] (Figure 4e). The Fe 2p spectrum of the PCBP-1 was curve-fitted into three components at 721.68, 712.18 and 708.48 eV corresponded to the low spin (Fe 2p_1/2_) states, the high spin (Fe 2p_3/2_) and the ferrocyanide of ferric ions [50], respectively (Figure 4g). The C 1s spectrum (Figure 4h) of the CBP was deconvoluted into three constituents: C–C (284.7 eV), C–O (285.9 eV), and O–C–O (286.6 eV). The C 1s spectrum (Figure 4i) of the PCBP-1 was curve-fitted into four components at 284.70, 285.02, 286.59 and 287.74 eV assigned to the C–C, C–N, C–O and O–C-O, respectively. The C–N peak proved the presence of PB in the PCBP, which was absent in the C 1s spectrum of CBP.

### 2.2. The Properties Analysis of PCBP

It is inevitable that any material will come into contact with various liquids in the course of its use. Therefore, the liquid resistance ability of materials would be urgently needed in certain cases. Nevertheless, compared to fossil-fuel plastics, applications of cellulose-based products are strongly hampered by limited resistance to penetration by liquids. Therefore, high liquid-resistant properties of cellulosic-based products, e.g., cellulose paper, cellulose aerogel and cellulose film, are highly desirable. The general strategy for cellulosic products to obtain liquid barrier properties is through the surface anchorage of barrier coatings. Structural reorganization of microfibers involving the combination of dissolution, regeneration and hot-pressing as an alternative to the use of barrier coatings was found to be very effective in developing liquid barrier properties. As shown in Figure 5a, CBP and PCBP all had a strong resistance to penetration by colored ethanol, grease solution and water. The water-contact angles for CBP and PCBP-2 were 60.6 and 73.3°, respectively—still less than 90° (Figure 5b,c).

The mechanical property is another critical parameter for the practical application of plastic materials. The mechanical strength of the photothermal bioplastics constructed from PB and cellulose support is a prerequisite for the long-term operation in a harsh environment. Figure 6a shows the stress–strain curves for CBP, PCBP-1 and PCBP-2. The CBP possesses a tensile strength of 85.7 MPa. The mechanical strengths of PCBP-1 and PCBP-2 were 77.2 MPa and 65.7 MPa, respectively (Figure 6b). Compared with CBP, the tensile strength of PCBPs decreased. The main reason for the decrease in the mechanical strength of PCBPs was the addition of hydrochloric acid during the formation of PB. In Figure 6a, the inserted digital image shows a good mechanical strength and flexibility of PCBPs. Meanwhile, Appendix A shows the tailoring process; note that no brittle failure occurred during tailoring of the CBP and PCBP samples. Furthermore, distinguished mechanical robustness of PCBP-2 was clearly identified. CBP and PCBP-2 were able to support a metal flattener (10 kg; Appendix A). Appendix A show the mechanical robustness of CBP and PCBP-2 in regard to the metal flattener.

### 2.3. Photothermal Conversion Behavior of PCBP and Applications

Next, the xenon lamp was used to simulate solar radiation to study the solar-to-thermal properties of PCBPs. The surface temperature of PCBPs under solar irradiation was monitored using an infrared camera. Figure 7a,c,d show that the surface of PCBPs was rapidly heated up, and reached 61.2 °C on PCPB-1 and 70.3 °C on PCBP-2 under standard one sun illumination. The temperature of the PCPB-1 and PCBP-2 increased from ~25 °C to 61.2 °C and ~70.3 °C during 6 min illumination, indicating that the bioplastic films have good light absorption and photothermal conversion performance. The surface temperature of PCBP-2 can be further increased up to 88.9 °C under two sun (200 mW cm^−2^) illumination (Figure 7b,e). The light absorption ability of PCBP at wavelengths ranging from 200 to 2400 nm was investigated. As displayed in Figure 7f, the solar reflectance of PCBP-2 demonstrated lower values than the controlled sample (pristine CBP). The overall reflectance of PCBP-2 was below 10% across the wavelength from 200 to 1200 nm. Meanwhile, the light absorption of PCBP-2 was significantly higher than that of pure CBP as shown in Appendix A. PCBP-2 showed a high light absorption capacity, especially in the visible and near-infrared wavelength range (200–1200 nm), which belongs to the main distribution region of solar energy. This result indicates PCBP-2 has strong absorption, which is attributed to the uniform distribution and the strong light absorption capacity of PB particles. As displayed in Figure 7g, the stable light-to-heat conversion capability of the PCBP-2 was proven by repeated heating/cooling cycles, including turning on the xenon lamp irradiation for 240 s, followed by turning it off for 160 s. The quick light-to-heat conversion capacity was well-maintained during four heating/cooling cycles.

Based on the excellent photothermal conversion of PCBPs, we next studied whether PCBPs can be used to drive Stirling engine motion. The Stirling engine (or engine) was first invented by brit Robert Sterling in 1816, and is an externally heated (or burnt) piston engine. The working medium is mainly gas, that is, a temperature difference is applied to both sides of the engine box, and the air in the engine box is reversibly compressed and expanded, thereby driving the reciprocating motion of the diaphragm and the continuous rotation of the engine turbine. PCBPs were fixed to a transparent substrate (such as a polymethyl methacrylate (PMMA) board) on the bottom side of the Stirling engine (Figure 8a). PCBPs absorb sunlight and heat the air at the bottom of the Stirling engine, generating a temperature difference between the top and bottom of the engine box. Figure 8b shows a physical photograph of the Stirling engine and non-contact speed meter. The rotational speed of the Stirling engine was measured by a non-contact tachometer. As a result, when the sunlight penetrated the transparent PMMA plate (Appendix A) and was absorbed by PCBPs, the rotation of the engine turbine was observed. For a comparison purpose, the Stirling engine placed CBP only on the PMMA board, and no rotation occurred during solar irradiation, verifying that the rotation was triggered by the photothermal conversion caused by PCBP-1 and PCBP-2 (Appendix A). PCBPs with different loading amounts of PB particles triggered rotation of the engine turbine at different speeds, ranging from 0 to 154 r/min for PCBP-1 and 195 r/min for PCBP-2 (Figure 8c). When the PCBP-2 was used to drive the Stirling engine motion, changes in the illumination level (0.5, 1, 1.5 and 2 standard units of solar radiation) also triggered different rotational velocities (Figure 8d). All of these results demonstrate that the fabricated solar-thermal material can be used to drive a Stirling engine, and the rotation speed of the Stirling engine can be tuned by changing the loading amount of photothermal material and by changing the light intensity.

## 3. Materials and Methods

### 3.1. Materials

Filter paper (FP) was obtained from Hangzhou Special Paper Industry Co., Ltd. (Hangzhou, China). Sodium ferrocyanide (Na_4_Fe(CN)_6_), Lithium hydroxide (LiOH) and urea were obtained from Macklin Biochemical Co., Ltd. (Shanghai, China), Tian Da Chemical Reagent Co., Ltd. (Zhangqiu, China) and Tian li Chemical Reagent Co., Ltd. (Tianjin, China), respectively. Glycerol and hydrochloric acid (HCl) (36.0–38.0%) were of analytical reagent grade and used without further purification.

### 3.2. Synthesis of Prussian Blue Powder

The synthesis of Prussian blue (PB) was as follows: 8 mmol of Na_4_Fe(CN)_6_·10H_2_O was solubilized in 200 mL distilled water and 6 mL concentrated hydrochloric acid was subsequently added. Then, the above solution was reacted at 60 °C for 6 h under stirring; during the reaction procedure, the mixture gradually turned blue from light yellow. The product was collected by centrifugation and washed three times with water and ethanol, then dried at 70 °C overnight to obtain the final PB powder.

### 3.3. Fabrication of Cellulose Hydrogel

Briefly, powder-like cellulose was added to LiOH/urea solution (4.6 wt%/15 wt%) and stirred for 5 min at room temperature. Then, the above mixture was frozen in the refrigerator at −20 °C for 12 h to form 4 wt% transparent Li–cellulose solution. The resultant viscous Li–cellulose solution was subjected to ice water bath sonication for 20 min to remove air bubbles. The viscous bubble-free Li–cellulose solution was cast on a glass mold, followed by anhydrous ethanol being used as the green non-solvent to induce rapid regeneration, and then thorough washing with distilled water to obtain cellulose hydrogel with about 2.0 mm of thickness.

### 3.4. Fabrication of Prussian Blue@Cellulose Bioplastics

In this process, cellulose hydrogel was immersed in a 200 mL solution containing 8 mmol Na_4_Fe(CN)_6_·10H_2_O for 6 h. Subsequently, 6 mL concentrated HCl was added. Then, the reaction system was heated to 60 °C and maintained and reacted for 6 h. The reaction process was repeated 2 times. PB@cellulose hydrogel was then removed from mixtures and washed with water to neutrality. Depending on the number of reactions, the PB@cellulose hydrogels were labeled as PB@cellulose hydrogel-1 and PB@cellulose hydrogel-2. The PB@cellulose hydrogels were immersed in 5 *w*/*w*% glycerin aqueous solution for 30 min. Before hot pressing, the composite cellulose hydrogels were dried at room temperature for 1 h to remove some of the moisture. The resulting PB@cellulose hydrogels were sandwiched between stainless steel plates and then dried at 110 °C at an applied pressure of about 0.1 MPa initially, and finally were hot pressed at an applied pressure of about 60 MPa with an R32022015 hot press machine (Schuler, Shanghai, China). Thus, photothermal Prussian blue@cellulose bioplastic films with a thickness of about 0.20 mm were obtained, called as PCBP-1 and PCBP-2. For comparison purposes, a pure cellulose bioplastic (CBP, 0.20 mm) was obtained using the same process.

### 3.5. Characterization

Light reflection of the samples was tested using a Cary 5000 UV–Vis–NIR spectrophotometer over the spectral range 200–2400 nm. The transmittance of neat CBP in the wavelength from 200 to 800 nm was tested. IR images were collected using a Testo 869 IR camera (Testo SE & Co. KGaA, Lenzkirch, Germany). A CEL-S500 xenon lamp (Aulight Co., Ltd., Beijing, China), which simulates solar light, was used as the light source. Fourier transform infrared (FTIR) spectra were collected on a FTIR spectrometer (Nicolet 6700, Thermo Fisher Scientific Inc., Waltham, MA, USA) in the frequency range of 4000–500 cm^−1^. The scanning electron microscope (SEM, Hitachi S4800, Hitachi High-Tech Corporation, Tokyo, Japan) was used to identify the surface and cross-sectional morphologies, and surface elemental composition analysis was conducted by energy-dispersive X-ray spectroscopy (EDS). The samples were frozen in liquid nitrogen, snapped immediately, and then used for SEM analysis. A D/max 2200PC X-ray diffractometer (Rigaku Corporation, Japan) with Cu Ka radiation (λ = 0.154 nm) was used to collect the X-ray diffraction (XRD) patterns. X-ray photoelectron spectroscopy (XPS) test was carried out with an ESCALAB 250Xi (Thermo Fisher Scientific, Waltham, MA, USA) using Al Kα radiation. Thermal gravimetric analysis (TGA) was performed by a STA 449 F3 (NETZSCH-Gerätebau GmbH, Selb, Germany) equipment under a nitrogen atmosphere at a heating rate of 10 °C·min^−1^. The tensile stress–strain curves were carried out at ambient temperature using a UTM2203 universal testing machine (Shenzhen SUNS Technology Stock Co. Ltd., Shenzhen, China). Rectangular strips with 5 mm × 50 mm dimensions were used in tensile tests. The water contact angles of samples were performed by a Data Physics Instrument (drop shape analysis system DSA-100/10, KRÜSS) in dynamic mode. Water, ethanol, acetone, toluene, DMAC, N-propanol and methanol were used to test the solvent resistance of bioplastic films. Pieces of photothermal bioplastic films with the size of 6 mm × 50 mm were immersed in solvent for 90 days, and they were photographed to show their solvent resistance performance. Simultaneously, CBP and PCBP-2, equipped as filters, were combined with glass filtration sets. Grease solution (mixture of castor oil, toluene, and n-heptane), colored ethanol and water were used to interact with CBP and PCBP for 30 days. Digital photographs were collected to evaluate liquid-barrier performance.

### 3.6. Light Drive Stirling Engine Motion Test

The tests of light drive Stirling engine motion were carried out under xenon lamp irradiation. The xenon lamp light was employed to simulate sunlight. For the light drive Stirling engine motion test, the iron support with the supporting iron ring was placed directly above the xenon lamp holder, then a piece of transparent polymethyl methacrylate (PMMA) was placed on the iron ring, and the photothermal bioplastics (PCBP-1 and PCBP-2) with a size of 6 cm × 6 cm were placed on the PMMA board. Finally, the Stirling engine was pressed onto the bioplastic film to make it in full contact with the PCBP. When solar light penetrated the high transparent PMMA plate, the temperature difference was generated between the top and bottom of the Stirling engine, rotation of the Stirling engine was observed. The rotational speed of the Stirling engine was measured by a non-contact speed meter (AR926, Wan Chuang Electronics Mfg. Co., Ltd., Dongguan, China). For comparison purposes, a pure CBP was used to drive the Stirling engine motion.

## 4. Conclusions

In this work, with the premise of using green processes in mind, PCBP with excellent photothermal conversion performance was prepared by a novel and sustainable processing method, i.e., cellulose liquefaction, rapid gelation, in situ synthesis and hot-press drying. The obtained PCBP-2 can effectively absorb sunlight and the surface can be heated up to 70.3 °C under one solar illumination (100 mW cm^−2^) during 6 min. The cyclic photothermal conversion experiment suggested the good photothermal stability of the PCBPs. As a demonstration of the practicality of PCBP, it was successfully used to drive a Stirling engine motion. The PCBPs possessed a strong resistance to penetration by aqueous/nonaqueous liquids. The water-contact angle of PCBP-2 was up to 73.3°. As-prepared PCBPs exhibited a high mechanical strength and thermal stability. Our work provides a new method to utilize cellulose and PB in the fabrication of photothermal bioplastic materials that can convert sunlight into mechanical motion. Since Stirling engines had great potential in the energy area and PCBP could be fabricated by a simple, green and low-cost method, this work is expected to provide an efficient pathway to sustainable and clean energy. This simple and green strategy would direct sustainable cellulose-based photothermal bioplastic toward diversified applications: photothermal layer for thermoelectric generator, agricultural films for soil mulching and photothermal antibacterial activity, among others.

## Data Availability

Not applicable.

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
