# Peer review of "Mechanically Strong, Liquid-Resistant Photothermal Bioplastic Constructed from Cellulose and Metal-Organic Framework for Light-Driven Mechanical Motion"

_molecules, 2021, doi:10.3390/molecules26154449_

Round 1

Reviewer 1 Report

This is an interesting contribution, generally quite sound in results. 

I've some suggestions concerning naming and presentation summarized in the file attached.

There is mainly one scientific criticism, but that can easily be corrected.

Reviewer 2 Report

This manuscript presents the mechanically strong, liquid-resistant photothermal bioplastic constructed from cellulose and metal-organic framework for light-driven mechanical motion. There are a few remarks that, I hope, can help the authors to improve the text:

  1. The language should be improved.
  2. There are a lot of typo errors in the manuscript.
  3. Authors wrote “… corresponding to the diffraction of the (110), (1Î0) and (200) faces …”. What is the second planes (1Î0)? Please, check the correction of this plane crystal (Bravais) lattices.
  4. Information about the structural characteristics of studied materials should be added (strain and residual stress).
  5. Absorption spectra should also be added before and after illumination.
  6. Did Authors determine any thermal parameters?
  7. The thickness of the sample should be added.
  8. Authors should read and cite the following articles: Solar Energy 203, 19 (2020); Materials Chemistry and Physics 223, 700 (2019).
  9. More interpretation of the results should be added.

Summarize, the manuscript is average, however, it could be published in Molecules but after minor revision and linguistic correction.
